# Effect of Cognitive Behavioral Intervention Combined with the Resilience Model to Decrease Depression and Anxiety Symptoms and Increase the Quality of Life in ESRD Patients Treated with Hemodialysis

**DOI:** 10.3390/ijerph20115981

**Published:** 2023-05-28

**Authors:** Cristina J. González-Flores, Guillermo Garcia-Garcia, Claudia Lerma, Rebeca María Elena Guzmán-Saldaña, Abel Lerma

**Affiliations:** 1Centro Universitario de la Cienega, University of Guadalajara, Ocotlán 47820, Mexico; cristina.gonzalez4243@academicos.udg.mx; 2Nephrology Department, Civil Hospital de Guadalajara Fray Antonio Alcalde, Guadalajara 44280, Mexico; ggarcia1952@gmail.com; 3Instituto Nacional de Cardiología Ignacio Chávez, México City 14080, Mexico; lermag@unam.mx; 4Institute of Health Sciences, Universidad Autónoma del Estado de Hidalgo, San Juan Tilcuautla 42160, Mexico; rguzman@uaeh.edu.mx

**Keywords:** depression, anxiety, psychological resilience, quality of life, cognitive behavioral intervention, cognitive distortions

## Abstract

The aim of this study was to compare the effect of cognitive behavioral intervention (CBI) combined with the resilience model (CBI + R) vs CBI alone on depression symptoms, anxiety symptoms, and quality of life of end-stage renal disease (ESRD) patients undergoing hemodialysis replacement therapy. Method: Fifty-three subjects were randomly assigned to one of two treatment groups. The control group (*n* = 25) was provided with treatment strategies based on a cognitive behavioral approach, while the experimental group (*n* = 28) were given the same techniques plus resilience model strategies. Five psychological instruments were applied: Beck Depression Inventory, Beck Anxiety Inventory, Mexican Resilience Scale, cognitive distortions scale, and the Kidney Disease related Quality of Life questionnaire. Participants were assessed at baseline (before treatment), eight weeks later (end of treatment), and four weeks after the end of treatment (follow up). The results were analyzed by ANOVA for repeated measures with a Bonferroni-adjusted test method, with *p* < 0.05 considered significant. Results: The experimental group had significant differences in total and somatic depression as well as differences in the dimensions of cognitive distortions and a significant increase in the dimensions of resilience. The control group had significant differences in all variables but showed lower scores in the evaluated times. Conclusions: The resilience model strengthens and enhances the effectiveness of the cognitive behavioral approach to reduce symptoms of depression and anxiety in patients with ESRD.

## 1. Introduction

Negative emotional states, specifically depression and anxiety, are common disorders present in patients with end-stage renal disease (ESRD) [1,2,3]. Replacement therapies such as hemodialysis are invasive and affect the quality of life of a patient with ERSD [4,5]. Increased mortality and an increase in the rate of hospitalizations are associated with depression and anxiety symptoms in this population [6,7]. Shulman and Spinelli found that the survival of hemodialysis patients with a score <14 on the Beck Inventory was 85%, while those with scores greater than 25 in the same instrument had a survival rate of 25% [8]. A study by Kellerman estimated that, for every point increase in the inventory of Beck, the mortality risk increased by 2.7% [4].

Some psychological models have implemented several strategies to reduce depression and anxiety [9,10,11]. However, cognitive behavioral therapy has shown stronger evidence of results in the treatment of these symptoms [12,13,14]. In the resilience model, the psychological resilience of the individual emerges as a resource in adverse situations [15]. Resilience is a construct that has been defined in a broad range of models and, because there is no shared definition, the concept of resilience is complex. However, resilience is much more than resistance to trauma—it expresses the ability to react positively despite difficulties, turning them into opportunities for growth [16]. Resilience consists of personalized skills to cope with adverse situations and to even emerge stronger from them. In chronic disease, resilience can be associated with adherence to treatment and well-being [17,18,19,20,21,22,23,24]. Although there is evidence that psychological resilience acts as a protective factor against depression and anxiety, resilience has scarcely been evaluated in ESRD patients [18,19,23,24,25,26,27,28]. The dimensions of resilience (i.e., strength, self-confidence, social competence, social support, family support, and self-structure) can be mediators in the reduction of negative emotional states in ESRD patients [20]. In chronic patients, resilience can be a modulator of depression and anxiety [23,29,30,31,32], and can serve as a therapeutic target for enhancing models whose results have shown evidence of effectiveness. Psychological resilience is considered an important factor for protecting mental health and is a moderating variable in depression and anxiety symptoms [32,33,34,35,36]. In ESRD patients, there are significant correlations between depression, anxiety, and resilience [20,32], which suggests that resilience may function as a protective factor against these symptoms. Resilience acts as a personality factor and promotes physical and mental health [32].

Among the many models for the treatment of depression and anxiety symptoms in hemodialysis patients [31], a brief cognitive behavioral intervention is effective in reducing depressive and anxiety symptoms and improving quality of life in patients with ESRD undergoing hemodialysis compared to the usual treatment [30]. However, whether this cognitive behavioral intervention improves effectiveness by including additional techniques from the resilience model has not been evaluated. The purpose of this study was to compare the effect of a cognitive behavioral intervention (CBI) combined with the resilience model (CBI + R) vs CBI alone on depression symptoms, anxiety symptoms, and quality of life of ESRD patients undergoing hemodialysis replacement therapy.

## 2. Materials and Methods

### 2.1. Patients and Study Design

The study involved 64 patients selected by random sampling of the nephrology department of the Hospital Civil de Guadalajara Fray Antonio Alcalde, Mexico. Patients were selected through a screening process conducted on an initial 151 patients whose symptoms of depression and anxiety were evaluated (Figure 1). Sixty-four patients with mild to moderate scores were included and randomly assigned to one treatment group. During treatment, a total of eleven patients were lost due to different reasons (i.e., received kidney transplant, required hospitalization, had personal problems, or died). Therefore, fifty-three patients received the complete treatment and follow up (25 in the CBI group and 28 in the CBI + R group).

Patients assigned to the control group had CBI while those assigned to the experimental group had the same CBI with two additional sessions of resilience intervention (CBI + R). Both groups attended hemodialysis therapy three times a week and continued the usual pharmacological and nutritional treatments. Only literate patients without psychiatric comorbidities and no hospitalizations in the last six months were included. After an explanation of the protocol, all participants signed the informed consent form. The study protocol was approved by two committees relating to Research and Bioethics (number CB/023/2017 and 298/17 protocol). Patients who were identified with severe symptoms of depression and/or anxiety or had suicidal ideation according to item 9 of the Beck Depression Inventory were referred to psychiatric services for assessment and treatment.

Three evaluations were carried out at different times for both groups. The first evaluation was performed at the baseline time (before the first treatment session). The second was performed at the end of treatment, thus eight weeks after the baseline measurement. The last (follow up) was completed four weeks after the end of treatment.

### 2.2. Design of the Intervention

The intervention program was CBI in the control group and CBI + R in the experimental group. Standardized handbooks were used for each intervention, one handbook for the patient and one for the therapist. Each handbook included information and instructions for each activity, exercise, and assigned task.

Interventions for the CBI and the CBI + R groups were similar except for the last two sessions of the experimental group, which individuals received the resilience model strategies, while the control group strengthened the content of the previous six sessions with the cognitive behavioral model only. The intervention in both groups was based on classic techniques from the cognitive behavioral model [37], as well as material previously utilized in diabetic [38] and kidney patients [30].

On the other hand, the sessions based on the resilience model were designed based on previously conducted programs belonging to the model Resilient [39,40]. The content of the handbooks was adapted to the current context using techniques and strategies suitable for the characteristics of the population. The content of the handbook included images, examples, explained exercises, and other elements related to ESRD patients. Each technique used different components. For example, in the cognitive behavioral model, for the session about cognitive restructuring, images with cartoons representing people having catastrophic thoughts were shown to all patients, and each patient used a Word format to write down their own thoughts related to ESRD. During the intervention, the patients were invited to talk about real examples in their daily lives. The techniques were explained and then practiced in each class when the patient had achieved a complete understanding of the material, then, this knowledge was reinforced through the weekly homework described in the handbooks. The intervention was designed for small groups with four participants for each group and one session per week. The participants in both groups (CBI and CBI + R) attended eight sessions lasting two hours each.

The therapist handbook was organized in each session with an introduction, goals for the session, a review of homework and self-completed forms, a description of the session material, and related exercises. On the other hand, the patients’ handbook contained the same topics but with the formats adapted for the patients to practice the exercises at home. Through a pilot test, the handbook was tested with 5 healthy volunteers. This process verified the session duration, the facility’s functionality, and the procedures. This verification process allowed us to identify and correct the intervention elements to ensure adherence to the designed intervention.

The following techniques were applied throughout the eight sessions of the CBI:

*Behavioral activation.* The technique sought to motivate the patient to undertake pleasurable activities that they had not undertaken due to their illness. This technique has shown effectiveness in counteracting depression by increasing activity levels and exposure to positive reinforcers [41]. Self-reinforcement was implemented to train patients with ESRD to maintain appropriate behavior triggered by positive stimuli [42].

*Cognitive restructuring*. This technique, which identifies negative cognitions that happen by distortion of thoughts, was provided [43,44].

### 2.3. Deep Breathing and Muscle Relaxation

The strategy consists of controlled breathing and exhalation, as a strategy for managing anxiety. The patients were taught muscle relaxation exercises for opposing muscle pairs (short periods of 5–7 s of sustained muscle contraction). The aim is to increase the level of well-being in the patient via muscle relaxation to physically reduce anxiety symptoms [45] and, in combination with diaphragmatic breathing, it leads to an emotional state of tranquility. These techniques have been used effectively in other populations to decrease anxiety symptoms [46].

The experimental group (CBI + R) used the same content to decrease depression and anxiety symptoms. Additionally, the CBI + R group had two sessions of the resilience model with the following content:

### 2.4. Self-Confidence and Social Support

Patients were invited to participate in the activity called “sharing my fears”, which was aimed at representing their fears related to the disease and to identify social support as a coping mechanism when facing such fears. Patients used creative materials (colored clay and crayons) for modeling figures and drawing cartoons or artwork to represent their fears. Then, patients were asked to share their own experience of fear and its related negative consequences, such as social isolation. The therapist leads the patients through questions to identify how social support was helpful to them in facing their fears and to restructure their distorted ideas about such fears. Through this activity, the patient externalized the situation and elaborated on a new meaning for his/her cognitive beliefs. The activity finished after each patient shared their learning with the group [39].

### 2.5. Self-Esteem and Gratitude

This activity was inspired by the theoretical model proposed by Kiswarday and Henderson [47], which consists of six resilience steps and proposes the promotion of resilience in educational contexts or formal institutions (such as the hospital and its hemodialysis unit). The model includes the promotion of assertiveness, life skills, problem-solving skills, and the ability to leverage resilience based on the affective resources of each person.

In this activity, the therapist introduced a drawing of a tree taped to the wall and called “the tree of life” and invited the patients to write short stories about adversity in relation to their experience of ESRD using one half of the tree. Then, in a group, they shared their emotions and thoughts about their expression and experience of the story.

In the second part of the activity, the patients wrote on the other half of the tree about how those experiences of adversity were lessons learned in their lives and which life skills they obtained from those circumstances. The activity promoted the resignification of the adversity story through acceptance, a new narrative, and gratitude towards the learning obtained in the adverse circumstance [39,40].

### 2.6. Sample Size Calculation

The sample size was calculated using a formula that compared mean values between two groups. Based on the mean Beck Depression Inventory (BDI) scores at the end of treatment in a previous study [48], the minimum difference between the intervention and control groups was 7.1, and the variance from the reference group (control) was 82.8. Considering an alpha error of 0.05 and a beta error of 0.20 (power = 80%), the number of patients was determined to be 20 per group.

### 2.7. Sociodemographic, Clinical, and Psychological Assessment

For each participant the following sociodemographic variables were evaluated by a data sheet that was collected as part of the study: age, gender, grade level, employment status, living status, number of dependents, medication use, comorbidities, previous transplant, smoking status before getting sick or at the time of the evaluation, and months on hemodialysis. The medical record of every patient was consulted by accessing the hospital database to obtain the latest results for biochemical variables: glucose (mg/dL), hemoglobin (mg/dL), albumin (g/dL), and creatinine (mg/dL).

All psychological questionaries were previously validated in the Spanish language. Depression symptoms were assessed with the BDI which has a reliability (Cronbach’s alpha, α) of 0.87 [30,49]. The BDI assesses two dimensions: (i) somatic symptoms (α = 0.86), with items such as “In the last two weeks I get more tired than usual”, and (ii) cognitive symptoms (α = 0.81), with items such as “I am critical of myself for my weaknesses or mistakes”. The cut-off points usually accepted to grade the intensity/severity of depression symptoms are: normal (0–9 points), mild (10–16 points), moderate (17–29 points), and severe (30–63 points) [30,49]. The anxiety symptoms were assessed with the Beck Anxiety Inventory (BAI) which has a Cronbach’s alpha of 0.90 [50]. The BAI has two dimensions: (i) somatic symptoms (α = 0.83), with items such as “Hands trembling”, and cognitive symptoms (α = 0.84), with items such as “Fear of worst happening”. This instrument is scored with whole numbers (0 to 3 points), it is categorized as normal anxiety (0 to 7), mild (8 to 15), moderate (16 to 25), and severe (26 and more), reaching a total score of 63 points. Both questionaries have 21 Likert-type items with a range of 0 to 3 points each and are used in both the general population and in ESRD patients [30,51,52,53].

Quality of life was evaluated by the KDQOL-36 scale developed by the Kidney Disease Quality of Life Working Group. It has 36 items divided into 5 dimensions, the scores range from 0 to 100 points and the internal consistency of the instrument is 0.80 [54]. The questionnaire has the following dimensions: (i) burden of the disease (α = 0.85), an item belonging to this subscale is “Your state of health limits you in doing activities”; (ii) symptoms and problems (α = 0.89), with items such as “During the last four weeks, have you felt muscle pain?”; (iii) effects of the disease (α = 0.85), with items such as “How much does the disease impact sexual life?”; (iv) a physical component dimension (α = 0.96), with items such as “He has felt exhausted and without strength”; and (v) a mental component (α = 0.86), with items such as “You have felt discouraged or sad” [54].

Resilience was evaluated using the Mexican Resilience Scale, which has 43 items divided into 6 dimensions with a global internal consistency of 0.93 [55]. The scale assesses a strength and self-confidence dimension (items 1–19, α = 0.93), social competence (items 20–27, α = 0.87), family support (items 28–33, α = 0.87), social support (items 34–38, α = 0.84), and structure (items 39–43, α = 0.79) [55].

Cognitive distortions were evaluated by the questionnaire via 30 items divided into 5 dimensions (catastrophizing, dichotomous thinking, outside self-worth, negative self-labeling, and perfectionism) and global internal consistency of 0.93 [56]. The scale has five dimensions: catastrophism (α = 0.88), dichotomous thinking (α = 0.84), intrinsic perfectionism (α = 0.80), extrinsic perfectionism (α = 0.78), and negative self-labeling (α = 0.76) [56].

### 2.8. Statistical Analysis

Kolmogorov–Smirnov tests were performed to test for a normal distribution in the continuous variables. Sociodemographic and clinical variables were compared between groups by *t*-tests (continuous variables) or X^2^ tests (categorical variables). Data are expressed as mean ± standard deviation or absolute value (percentage). Each psychological variable was compared using ANOVA for repeated measures with post hoc tests adjusted by the Bonferroni method with comparisons between groups by treatment (controls vs experiment) and comparisons within groups by time (baseline measurement, end of treatment, and follow up). For the total score of each questionnaire and for each sub-scale a separate ANOVA was applied. Statistical analysis was performed using the computer program SPSS version 21.0 (IBM Corp, Armonk, NY, USA).

## 3. Results

The characteristics of the study participants are shown in Table 1. There were no significant differences in any characteristics between the groups, except for a higher proportion of patients awaiting transplant in the CBI + R group compared with the CBI group.

Table 2 shows the scores for symptoms of anxiety and depression. Compared to baseline in both groups, the scores for both anxiety and depression had decreased at the end of treatment and remained low during follow up. This effect was observed both in the total symptom scores and in the somatic and cognitive subscales. In total, the for the anxiety subscale there were no differences between groups for the three measurement times; decreasing symptoms were similar between the two therapies. Moreover, the group R + CBI had lower total depression symptoms and somatic depression symptoms at the end of treatment compared to the group CBI. This difference was not maintained during follow up. For the cognitive subscale of depression there were no significant differences between groups at the three measurement times.

Comparisons of the quality-of-life scores are shown in Table 3. The quality-of-life scores increased after treatment in both groups. The score for the overall quality of life in the group CBI + R maintained a steady increase after treatment. In all dimensions of the instrument, the perception of quality of life increased in the CBI + R group, except for the physical component, which showed the greatest increase during follow up in the CBI group. Comparisons between groups showed similar values in both treatment groups.

The scores for cognitive distortions are shown in Table 4. Compared with the baseline scores, cognitive distortion scores had decreased at the end of treatment in both groups and all dimensions, except catastrophism in group CBI. During follow up, scores remained lower than baseline in the overall rate of distortion and dichotomous thoughts. The other subscale scores remained low compared to baseline only in the group CBI + R (extrinsic perfectionism self-labeling, total perfectionism, and negative self-labeling), while for the CBI group, scores during follow up were similar to baseline. Extrinsic perfectionism did not maintain low scores during follow up compared with the baseline in both groups. The negative self–labeling in the CBI + R group increased during the follow up compared to the end of the treatment period but retained low scores compared to baseline. In comparisons between groups, similarity was observed in all dimensions at the three follow-up times, except for intrinsic perfectionism, total perfectionism, and negative self–labeling, which had lower scores in the CBI + R group compared to the CBI group at the end of treatment.

Table 5 shows the resilience scores of the study participants. In the group CBI + R, total resilience scores increased in all dimensions at the end of treatment, and remained high during follow up compared with baseline scores. In contrast, in the group CBI, resilience scores at the end of treatment increased from baseline only for strength and self-confidence and structure, while in all other dimensions scores increased during the follow up compared to baseline. Only in the CBI group did the social competence score increase during follow up compared to the end of treatment. Resiliency scores of the CBI + R group were higher than those of the CBI group evaluated at the end of treatment for all dimensions except social competence. Both groups had similar scores on all dimensions of impact strengths at baseline and follow-up evaluations.

## 4. Discussion

### 4.1. Main Contribution

The main contribution of this work is in showing that, compared with CBI alone, the addition of salutogenic strategies from the resilience model to a CBI (CBI + R) leads to a similar decrease in both depression and anxiety symptoms and a similar increase in perceived quality of life in ESRD patients. Furthermore, in addition to improving the perception of resilience resources in the face of experience of the disease, the combined intervention (CBI + R) facilitated cognitive restructuring and the redefinition of adverse experiences from a positive (resilience) point of view.

### 4.2. Comparison with other Interventions

We found that the CBI group results coincide with those reported in the literature regarding the effectiveness of the cognitive behavioral approach to depression and anxiety symptoms in patients with medical comorbidities [30,41,57]. The results of this study show that the combination of the cognitive behavioral model, which evidence has shown to be effective in the reduction of depression and anxiety symptoms [30,58], together with the resilience model, which has also had a positive impact on the quality of life and reduction of negative emotional states [59], is a useful tool for addressing depression and anxiety symptoms in patients. This perspective diversifies psychological interventions in patients with ESRD and is committed to the promotion of salutogenic variables as an intervention tool in medically ill populations [59,60,61].

### 4.3. Theoretical Framework of the Resilience Contribution to the CBI

ESRD is a paradigmatic model of high risk for depression and anxiety when the patient’s resources are surpassed by the many stressors related to this illness. The interactions between psychosocial risk factors for disease and medical aspects of ESRD represent an opportunity for the study of interventions in chronic illness.

The psychological impact and overload of diseases justify greater emphasis on variables that allow a means of positively coping with situations of adversity, where resilience is a vital element in the psychological treatment of the medically ill population. Resilience is a protective factor in reducing negative symptoms such as depression and anxiety, acting as a personality factor and promoting mental health. The study and immersion of the construct in other disciplines are of interest in various areas such as the social and health sciences [32].

Resilience can function as a modulator against the symptoms of depression and anxiety [23,29,32], it is a protective factor in the mental health process of chronic patients [32,36], in addition to promoting coping strategies in situations of adversity [31]. It is for these reasons that interventions that include the resilience model will make it possible to work on aspects of resilience such as strength, self-confidence, social competence, social support, family support, and self-structure. In this study, the intervention consisted of managing and understanding these dimensions of resilience. Compared to CBI alone, CBI + R increased the resilience scores, particularly in the dimensions of strength and self-confidence, social competence, and family support. The patients learned to reconstruct stories of adversity, obtaining a new meaning from them, which increases their level of confidence and may improve their self-esteem. They also learned to identify support networks and socialization strategies. The mere act of sharing their emotions, thoughts, and behaviors promoted a degree of self-confidence in the patient groups [40]. This approach allows a dynamic interaction between peers in which each patient can discover his/her individual path to increasing resilience through experiences shared by the other patients. Since each person faces adversity in a different way, the dynamic interaction promoted in our intervention is likely to encourage the strengthening of more resilience factors than the few dimensions assessed in the resilience questionnaire, in agreement with the framework of flexibility sequence (or flexible self-regulation) discussed by George A. Bonanno [62].

### 4.4. Clinical Implications

The implementation of psychological interventions in medically ill populations represents an alternative to comprehensively improving the welfare of this population [9,63], which may directly impact the institutional costs [64,65,66]. The present study shows that interventions aimed at reducing mild-to-moderate symptoms are a timely and effective strategy in mental health services for providing care to at-risk populations such as ESRD patients. The technique of cognitive behavioral therapy is a reliable and replicable model due to its systematic and brief nature. The results of the study can be integrated into other models of mental health care, such as the model called stepped care, which aims to ensure that people have optimized access to the appropriate services for their needs over time [67,68].

Total somatic depression showed significant differences at the end of treatment compared to baseline measurement in the CBI + R group. Some studies have addressed the cognitive behavioral approach combined with salutogenic models like mindfulness [53,69], coping [70], and social support [71]. Besides decreasing symptoms of depression and anxiety perception for quality of life, the higher scores in the resilience dimensions of the group CBI + R show that resilience acts as a protective factor and increases the perceived quality of life in populations with experiences of adversity [72,73,74]. Total perfectionism and intrinsic and negative self-labeling significantly differed at the end of treatment and decreased versus the baseline. These results are consistent with the literature on cognitive distortion [30].

As for the variable resilience, the highest score was obtained for all of its dimensions in the group CBI + R. Some studies have raised the importance of integrating resilience as a central factor in psychological interventions [75]. The results generated in this study suggest important clinical implications for future interventions. The resilience model techniques play a modulatory role in reducing depression and anxiety symptoms through cognitive behavioral-based interventions. Some research has used resilience as a modulatory factor with different variables and with positive results [76,77]. The inclusion of the resilience model within the CBI preserves the effectiveness of the cognitive behavioral approach to treating mild and moderate depression and anxiety symptoms while improving the positive perception of the patients‘ own resources.

### 4.5. Study Limitations

In the experimental group, six sessions of the cognitive behavioral model and two sessions based on the resilience model were applied. In contrast, the control group received six sessions of the cognitive behavioral model plus two reinforcement sessions ( completing eight sessions). Although this ensured that the intervention total duration was the same in both groups (i.e., 8 weeks), the exposure to the cognitive behavioral model was not identical between groups since the experimental group had no reinforcement sessions.

## 5. Conclusions

Compared to CBI alone, the resilience model (CBI + R) has a similar clinical effect on decreasing depression and anxiety symptoms while increasing the quality-of-life perception in ESRD patients undergoing hemodialysis replacement therapy. The CBI + R group did better than the CBI group on the cognitive restructuring outcomes (i.e., cognitive distortion scores). Furthermore, the resilience model is a novel and strategic way to integrate salutogenic variables into a psychological intervention that promotes positive aspects and mediating resources to cope with stressors in the face of disease.

## Figures and Tables

**Figure 1 ijerph-20-05981-f001:**
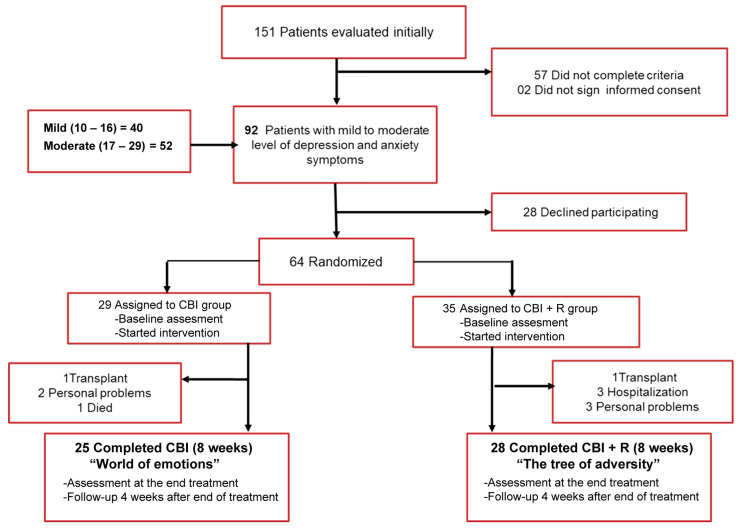
Flow chart of the screening process for recruitment of study participants.

**Table 1 ijerph-20-05981-t001:** Characteristics of the study participants. Data are shown as absolute value (percentage) or mean ± standard deviation.

Variables	Total (*n* = 53)	CBI + R (*n* = 25)	CBI (*n* = 28)	*p*
Age (years)	34 ± 12	34 ± 12	35 ± 13	0.70
Sex				0.33
Female	16 (31)	7 (25)	9 (38)	
Male	36 (69)	21 (75)	15 (62)	
Schooling				0.06
Primary school	38 (59)	27 (77)	16 (55)	
Secondary school or higher	18 (29)	8 (23)	13 (44)	
Living status				0.89
Alone	2 (3.1)	1 (2.9)	1 (3.4)	
With companion	62 (97)	34 (97)	28 (96)	
Employed	16 (25)	11 (31)	5 (17)	0.25
With dependents	12 (19)	6 (17)	6 (20)	0.71
On medications	41 (64)	23 (23)	18 (62)	0.76
With other comorbidities	42 (66)	24 (69)	18 (62)	0.58
Previous kidney transplant	5 (8)	2 (6)	3 (10)	0.49
Awaiting kidney transplant	24 (38)	17 (49)	7 (24)	0.04
Smoking	4 (6)	1 (2)	3 (10)	0.21
Past smoker	21 (33)	11 (31)	11 (38)	0.79
Dialysis vintage (months)	52 ± 47	58 ± 53	45 ± 38	0.31
Blood hemoglobin (mg/dL)	8.5 ± 1.1	8.6 ± 1.1	8.5 ± 1.2	0.72
Serum albumin (g/dL)	3.5 ± 0.3	3.5 ± 0.3	3.5 ± 0.4	0.80
Serum creatinine (mg/dL)	7.6 ± 1.1	7.4 ± 1.2	7.9 ± 0.8	0.15
Serum glucose (mg/dL)	99.5 ± 16.6	102.9 ± 18.7	95.6 ± 12.9	0.11

**Table 2 ijerph-20-05981-t002:** Scores for anxiety and depression symptoms. Data are shown as mean ± standard deviation.

	CBI + R (*n* = 25)	CBI (*n* = 28)	*p*
Anxiety total score			
Baseline	16.39 ± 6.44	15.21 ± 9.08	0.25
End of treatment	5.04 ± 3.49 ^&^	5.48 ± 4.96 ^&^	0.80
Follow up	5.50 ± 3.89 ^&^	6.91 ± 4.98 ^&^	0.19
Somatic anxiety score			
Baseline	11.97 ± 5.09	10.25 ± 6.95	0.07
End of treatment	2.82 ± 2.84 ^&^	3.32 ± 3.71 ^&^	0.67
Follow up	4.29 ± 2.81 ^&^	5.26 ± 3.87 ^&^	0.39
Cognitive anxiety score			
Baseline	4.91 ± 2.54	5.03 ± 2.80	0.89
End of treatment	2.21 ± 1.44 ^&^	2.16 ± 1.72 ^&^	0.96
Follow up	1.57 ± 1.75 ^&^	1.92 ± 1.66 ^&^	0.32
Depression total score			
Baseline	18.88 ± 5.49	17.48 ± 5.71	0.29
End of treatment	4.93 ± 3.91 ^&^	7.44 ± 3.38 ^&^	0.02
Follow up	5.54 ± 3.12 ^&^	5.75 ± 2.93 ^&^	0.82
Somatic depression score			
Baseline	11.00 ± 3.84	10.72 ± 3.90	0.75
End of treatment	3.48 ± 2.83 ^&^	5.40 ± 2.69 ^&^	0.02
Follow up	3.43 ± 2.33 ^&^	4.13 ± 2.07 ^&^	0.28
Cognitive depression score			
Baseline	7.59 ± 2.09	6.76 ± 2.26	0.11
End of treatment	1.96 ± 1.66 ^&^	2.80 ± 2.81 ^&^	0.16
Follow up	2.11 ± 1.96 ^&^	1.63 ± 1.43 ^&^	0.25

^&^ *p* < 0.01 compared to baseline measurement within the same group.

**Table 3 ijerph-20-05981-t003:** Quality-of-life scores. Data are shown as mean ± standard deviation.

	CBI + R (*n* = 25)	CBI (*n* = 28)	*p*
Overall quality of life			
Baseline	66.67 ± 13.59	65.71.81 ± 13.76	0.10
End of treatment	82.72 ± 7.66 ^&^	82.08 ± 7.35 ^&^	0.65
Follow up	85.10 ± 8.12 ^&^	82.67 ± 11.12 ^&^	0.37
Disease burden			
Baseline	36.38 ± 20.41	37.76 ± 26.99	0.83
End of treatment	75.44 ± 21.44 ^&^	66.40 ± 20.08 ^&^	0.12
Follow up	65.84 ± 20.34 ^&^	65.10 ± 21.95 ^&^	0.90
Symptoms and Problems			
Baseline	79.16 ± 13.41	78.08 ± 15.77	0.79
End of treatment	85.64 ± 11.83	87.41 ± 9.10 ^&^	0.55
Follow up	89.73 ± 10.15 ^&^	85.24 ± 14.47	0.19
Effects of disease			
Baseline	62.94 ± 20.51	61.72 ± 21.98	0.83
End of treatment	79.57 ± 15.74 ^&^	75.65 ± 20.20 ^&^	0.43
Follow up	90.96 ± 12.45 ^&,¶^	85.67 ± 19.28 ^&^	0.24
Physical component			
Baseline	46.40 ± 8.6	32.71 ± 3.05	0.25
End of treatment	52.46 ± 3.59 ^&^	52.76 ± 2.55 ^&^	0.67
Follow up	49.38 ± 5.02 ^&,¶^	49.98 ± 4.03 ^&^	0.74
Mental component			
Baseline	43.80 ± 5.4	40.93 ± 7.8	0.19
End of treatment	51.66 ± 5.06 ^&^	52.45 ± 4.55 ^&^	0.56
Follow up	53.91 ± 4.32 ^&^	52.55 ± 4.88 ^&^	0.35

^&^ *p* < 0.01 compared to baseline measurement within the same group. ^¶^ *p* < 0.01 compared to end of treatment within the same group.

**Table 4 ijerph-20-05981-t004:** Cognitive distortion scores. Data are shown as mean ± standard deviation.

	CBI + R (*n* = 25)	CBI (*n* = 28)	*p*
Intrinsic perfectionism			
Baseline	10.76 ± 3.26	11.34 ± 3.32	0.49
End of treatment	7.4 ± 2.38 ^&^	8.16 ± 1.72 ^&^	0.03
Follow up	8.61 ± 2.61	8.88 ± 3.08	0.71
Extrinsic perfectionism			
Baseline	4.23 ± 1.83	14.17 ± 1.69	0.66
End of treatment	2.71 ± 1.15 ^&^	2.92 ± 1.03 ^&^	0.43
Follow up	3.4 ± 1.34 ^&^	3.21 ± 1.06	0.61
Total perfectionism			
Baseline	14.88 ± 4.61	15.52 ± 4.47	0.66
End of treatment	9.75 ± 3.12 ^&^	11.08 ± 2.54 ^&^	0.03
Follow up	11.64 ± 3.87 ^&^	12.08 ± 3.96	0.63
Catastrophism			
Baseline	27.91 ± 10.96	25.10 ± 11.80	0.27
End of treatment	18.21 ± 7.48 ^&^	19.68 ± 6.90	0.46
Follow up	15.93 ± 5.32 ^&^	15.83 ± 3.77 &	0.94
Negative self -labeling			
Baseline	10.66 ± 3.48	9.76 ± 3.22	0.32
End of treatment	5.75 ± 1.5 ^&^	6.96 ± 2.03 ^&^	0.03
Follow up	7.71 ± 2.37 ^&,¶^	8.04 ± 2.42	0.62
Dichotomous thinking			
Baseline	15.40 ± 4.90	15.00 ± 6.04	0.81
End of treatment	9.86 ± 3.15 ^&^	10.48 ± 3.13 ^&^	0.64
Follow up	10.46 ± 3.49 ^&^	9.42 ± 2.74 ^&^	0.24
Total score			
Baseline	68.47 ± 19.76	65.36 ± 21.36	0.53
End of treatment	43.57 ± 12.42 ^&^	48.20 ± 10.22 ^&^	0.15
Follow up	47.25 ± 13.43 ^&^	46.66 ± 10.40 ^&^	0.96

^&^ *p* < 0.01 compared to baseline measurement in the same group. ^¶^ *p* < 0.01 compared to treatment for measurement in the same group.

**Table 5 ijerph-20-05981-t005:** Resilience scores. Data are shown as mean ± standard deviation.

	CBI + R (*n* = 25)	CBI (*n* = 28)	*p*
Total resilience score			
Baseline	127.31 ± 21.85	128.54 ± 18.73	0.65
End of treatment	147.10 ± 11.11 ^&^	133.66 ± 13.84	0.01
Follow up	140.76 ± 14.21 ^&^	145.12 ± 11.92 ^&^	0.34
Strength and self-confidence			
Baseline	51.59 ± 10.56	51.66 ± 8.57	0.69
End of treatment	63.25 ± 5.89 ^&^	58.04 ± 6.21 ^&^	0.01
Follow up	61.00 ± 6.88 ^&^	61.17 ± 6.55 ^&^	0.85
Social competence			
Baseline	22.37 ± 5.11	23.90 ± 3.79	0.29
End of treatment	27.82 ± 2.62 ^&^	23.88 ± 4.13	0.01
Follow up	26.04 ± 4.13 ^&^	28.04 ± 3.31 ^&,¶^	0.06
Family support			
Baseline	22.37 ± 5.11	23.90 ± 3.79	0.57
End of treatment	20.50 ± 2.78 ^&^	18.76 ± 2.58	0.03
Follow up	20.30 ± 2.86 ^&^	20.38 ± 2.58 ^&^	0.81
Social support			
Baseline	14.40 ± 2.69	15.17 ± 2.86	0.45
End of treatment	17.64 ± 2.29 ^&^	16.44 ± 2.27	0.05
Follow up	16.67 ± 2.49 ^&^	17.88 ± 2.15 ^&^	0.07
Structure			
Baseline	14.29 ± 3.25	14.03 ± 3.822	0.61
End of treatment	17.89 ± 2.34 ^&^	16.28 ± 2.18 ^&^	0.01
Follow up	17.42 ± 2.54 ^&^	17.67 ± 2.47 ^&^	0.63

^&^ *p* < 0.01 compared to baseline measurement in the same group. ^¶^ *p* < 0.01 compared to treatment for measurement in the same group.

## Data Availability

The data presented in this study are available on request from the corresponding author.

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
