# Peer review of "Effect of Cognitive Behavioral Intervention Combined with the Resilience Model to Decrease Depression and Anxiety Symptoms and Increase the Quality of Life in ESRD Patients Treated with Hemodialysis"

_ijerph, 2023, doi:10.3390/ijerph20115981_

Round 1

Reviewer 1 Report

This paper presents findings from a randomized prospective study comparing a resiliency-based cognitive-behavioral therapy with traditional cognitive-behavioral therapy among end stage renal disease patients. Strengths of this study include its use of a randomized controlled trial design and a robust set of assessments. These results have significant potential for application in a patient population with substantial unmet psychological needs. However, there are some additional elements that need to be addressed in the current draft.

  1. More details regarding the assessments reported as outcomes in tables 3, 4, and 5 (anxiety/depression, QoL, and cognitive distortion) are needed in the measures section. The text reports the overall scale lengths and reliabilities for these measures, but not the subscale lengths and reliabilties – however, it is the subscales that you are analyzing, so it is necessary to understand the psychometric properties of the subscales as well. It would also be helpful to include sample items for each subscale.

  2. It is stated that the ANOVA results are corrected for multiple comparisons, but it is not clear from the context whether this means that each individual analysis is Bonferroni-corrected (so p/3), or whether the entire list of all analyses in the manuscript is Bonferroni-corrected (so p/~45). I think probably the first, but it’s not clear. Although a more stringent approach might be preferable, given the large number of comparisons, I think it can be justified as it (assuming it is the less stringent method that it seems to be), but it needs to be explained in somewhat more detail.

  3. It is not clear when the follow-up was conducted, or what the follow-up consisted of. It does not seem to be explained in the methods at all.

  4. The first part of the conclusions section could be written more clearly to make it explicit that the CBI+R group did better than the CBI group on the cognitive restructuring outcomes.

  5. The abstract needs to be edited for clarity and for English grammar and fluency (the rest of the manuscript is written in fluent English).

The abstract needs to be edited moderately for English fluency and clarity. The rest of the manuscript is written in clear and fluent English.

Author Response

Comment: This paper presents findings from a randomized prospective study comparing a resiliency-based cognitive-behavioral therapy with traditional cognitive-behavioral therapy among end stage renal disease patients. Strengths of this study include its use of a randomized controlled trial design and a robust set of assessments. These results have significant potential for application in a patient population with substantial unmet psychological needs. However, there are some additional elements that need to be addressed in the current draft.

  1. More details regarding the assessments reported as outcomes in tables 3, 4, and 5 (anxiety/depression, QoL, and cognitive distortion) are needed in the measures section. The text reports the overall scale lengths and reliabilities for these measures, but not the subscale lengths and reliabilties – however, it is the subscales that you are analyzing, so it is necessary to understand the psychometric properties of the subscales as well. It would also be helpful to include sample items for each subscale. (Pag. 5

Response: We appreciate the Reviewer’s comments which helped to improve the quality of our manuscript. Additional details regarding the subscales of the psychological questionnaires were added on page 6 (paragraphs 2 to 5).

Comment: 2. It is stated that the ANOVA results are corrected for multiple comparisons, but it is not clear from the context whether this means that each individual analysis is Bonferroni-corrected (so p/3), or whether the entire list of all analyses in the manuscript is Bonferroni-corrected (so p/~45). I think probably the first, but it’s not clear. Although a more stringent approach might be preferable, given the large number of comparisons, I think it can be justified as it (assuming it is the less stringent method that it seems to be), but it needs to be explained in somewhat more detail.

Response: The ANOVA was performed on each individual variable, and therefore the correction by the Bonferroni method was applied to each individual analysis. This is now clarified in the revised manuscript (lines 271 to 276): “Each psychological variable was compared using ANOVA for repeated measures with post-hoc tests adjusted by the Bonferroni method with comparisons between groups by treatment (controls vs experiment) and comparisons within groups by time (baseline measurement, end of treatment, and follow-up). For the total score of each questionnaire and for each sub-scale a separate ANOVA was applied.”

Comment: 3. It is not clear when the follow-up was conducted, or what the follow-up consisted of. It does not seem to be explained in the methods at all.

Response: The follow-up was conducted 4 weeks after the end of treatment. The description of the follow-up was improved (lines 92 to 95 and Figure 1).

Comment: 4. The first part of the conclusions section could be written more clearly to make it explicit that the CBI+R group did better than the CBI group on the cognitive restructuring outcomes.

Response: Thank you for this suggestion. We updated the conclusion section accordingly (lines 433 to 435).

Comment: 5. The abstract needs to be edited for clarity and for English grammar and fluency (the rest of the manuscript is written in fluent English).

Response: We edited the Abstract for clarity as follows:

Abstract: The aim of this study was to compare the effect of cognitive behavioral intervention (CBI) combined with the resilient model (CBI + R) vs CBI alone on depression symptoms, anxiety symptoms, and quality of life of end-stage renal disease (ESRD) patients undergoing hemodialysis replacement therapy. Method: Fifty-three subjects were randomly assigned to one of two treatment groups. The control group (n = 25) was provided with treatment strategies based on cognitive behavioral approach, while the experimental group (n = 28) were given the same techniques plus resilient model strategies. Five psychological instruments were applied: Beck Depression Inventory, Beck Anxiety Inventory, Mexican Resilience Scale, cognitive distortions scale, and the Kidney Disease related Quality of Life questionnaire. Participants were assessed at baseline (before treatment), eight weeks later (end of treatment), and four weeks after the end of treatment (follow-up). The results were analyzed by ANOVAs for repeated measures with Bonferroni-adjusted test method, considering significant p < 0.05. Results: The experimental group had significant differences in total and somatic depression as well as differences in the dimensions of cognitive distortions and a significant increase in the dimensions of resilience. The control group had significant differences in all variables but showed lower scores in the evaluated times. Conclusions: Resilient model strengthens and enhances the effectiveness of cognitive behavioral approach to reduce symptoms of depression and anxiety in patients with ESRD.

Reviewer 2 Report

Thank you for sharing this interesting paper comparing  CBI + R and CBI alone to improve depressive symptoms, anxiety symptoms, quality of life. The article is well written  and the topic is interesting. However, I have a few minor observations to make:

-        the use of a self-administered scale to assess resilience can lead to biases that should be explored in the light of previous work as the following:  Bonanno GA. The resilience paradox. Eur J Psychotraumatol. 2021 Jun 30;12(1):1942642. doi: 10.1080/20008198.2021.1942642. PMID: 34262670; PMCID: PMC8253174

-       In light of the previous point, a question that comes to the reader’s mind is how resilience is helpful in patients if depressive and anxiety symptoms are not better improved in those patients with better scores at the resilient scale. Perhaps more emphasis could be placed on  the effect that the CBI+R has on some particular aspects evaluated by the resilient scale (e.g. strength and self-confidence, social competence and support).

-       In light of the previous points and considering that the results found no differences in the improvement of depressive and anxiety symptoms between the two applied approaches, I would think about changing the title.

-       Inclusion criteria can be explained more clearly (e.g., in the diagram, patients who receive transplant seem to be excluded from the study, but this choice is not explained in the text)

Author Response

Comment: Thank you for sharing this interesting paper comparing  CBI + R and CBI alone to improve depressive symptoms, anxiety symptoms, quality of life. The article is well written and the topic is interesting. However, I have a few minor observations to make:

-        the use of a self-administered scale to assess resilience can lead to biases that should be explored in the light of previous work as the following:  Bonanno GA. The resilience paradox. Eur J Psychotraumatol. 2021 Jun 30;12(1):1942642. doi: 10.1080/20008198.2021.1942642. PMID: 34262670; PMCID: PMC8253174

Response: We appreciate the Reviewer’s constructive comments, which helped to improve the clarity of our manuscript. Now the revised manuscript discusses the suggested reference (lines 383 to 390).

Comment: - In light of the previous point, a question that comes to the reader’s mind is how resilience is helpful in patients if depressive and anxiety symptoms are not better improved in those patients with better scores at the resilient scale. Perhaps more emphasis could be placed on  the effect that the CBI+R has on some particular aspects evaluated by the resilient scale (e.g., strength and self-confidence, social competence and support).

Response: The revised manuscript now mentions the effect of CBI + R on several resilience dimensions (lines 377 to 379).

Comment: -       In light of the previous points and considering that the results found no differences in the improvement of depressive and anxiety symptoms between the two applied approaches, I would think about changing the title.

Response: Our evidence shows that CBI + R performed better than CBI for the total score of depression as well as the somatic depression symptoms. Although we did not find a larger effect of CBI + R on the cognitive depression symptoms, all sub-scales of anxiety, and all sub-scales of quality of life, we prefer the current title considering that these three variables (depression, anxiety, and quality of life) have the most clinical relevance and are traditionally considered outcome variables. In contrast, cognitive distortion and resilience dimensions are considered as intermediate (or modulatory) variables, as we discussed in a previous work (Reference 32).

Comment: -       Inclusion criteria can be explained more clearly (e.g., in the diagram, patients who receive transplant seem to be excluded from the study, but this choice is not explained in the text)

Response: The revised manuscript describes the loss of patients that started treatment but did not finish the 8 weeks of intervention due to different reasons including kidney transplantation (lines 92 to 94). Also, the flow chart in Figure 1 was updated for clarity.
